# Socio-Economic Security of the Region in the Context of Human Capital Development

**Nataliya V. Yakovenko** [1,*] **, Lyudmila Semenova** [2] **, Marina Ye. Tsoy** [3] **, Galina I. Zavyalova** [4] **, Elena A. Semenova** [5] **and Irina A. Belenok** [6]

1   Directorate of the Research Institute of Innovative Technologies and the Forestry Complex, Voronezh State University of Forestry and Technologies Named after G.F. Morozov, 8 Timiryazev Str., 394087 Voronezh, Russia

2   Higher School of Hospitality Educational and Scientific Cluster, Institute of Management and Territorial Development, Immanuel Kant Baltic Federal University, 14 A. Nevsky Str., 236041 Kaliningrad, Russia

3   Department of Marketing and Service, Business Faculty, Novosibirsk State Technical University, 20 Karl Marx Prospect, 630092 Novosibirsk, Russia

4   Department of Philosophy, Cultural Studies and Sociology, Institute of Social and Humanitarian Innovations and Mass Media, Orenburg State University, 13 Pobedy Ave., 460018 Orenburg, Russia

5   Faculty of Innovative Engineering and Hospitality Technologies, Institute of Service, Tourism and Design, Pyatigorsk (Branch) of NCFU, North-Caucasus Federal University, 56 A Ukrainskaya Str., 357502 Pyatigorsk, Russia

6   ANEO APE "Siberian Academy of Tourism", 12/1 Spartak, 630007 Novosibirsk, Russia

*   Correspondence: n.v.yakovenko71@gmail.com

**Abstract:** Socio-economic security is a highly critical issue that is addressed by every country in the world. In order to counteract challenges and threats faced by Russia, the Economic Security Strategy of the Russian Federation has been formulated. The development of human capital was proclaimed as one of the key areas to be targeted by the state policy. In this regard, the primary task is to identify destabilizing factors of socio-economic security in the development of human capital, which determined the relevance of the article. The article analyzes the research approaches to defining the key features of socio-economic security and human capital. It highlights the main principles of the theory of human capital on the basis of which a systematic approach to the study of human capital is justified. The proposed approach aims to identify negative factors that weaken regional security. By adopting this approach to the data in hand, the socio-economic security of the Voronezh Region, the old-developed region of the Central Black Earth Region in Russia, has been assessed. The data used for the analysis were collected from the Federal State Statistics Service. The mapping of the results of socio-economic differentiation was carried out by methods of GIS-technology, in particular Quantum GIS 12. For the processing of the initial information, the of multivariate statistical and econometric analysis SPSS Statistic 20.0 was used. As a result, factors that will have a negative impact on the development of human capital in the foreseeable future have been identified, namely the deterioration of the demographic situation, in particular the aging of the population and its natural decline; the decline in the quality of education, imbalances in the labor market, declining living standards, and deteriorating public health. It is concluded that to overcome the critical situation in the sphere of human capital and to ensure socio-economic security, it is necessary to carry out preventive measures.

**Keywords:** socio-economic security; human capital; Voronezh region

## 1. Introduction

The achievement of a high level of socio-economic security is one of the priorities of the National Security Strategy of the Russian Federation.

However, to date, the modern development of the national economy has been characterized as unstable for several reasons.

According the to official statistics, despite the development of the domestic sector of knowledge-intensive industries and the annual increase in spending on the development of the digital economy, which affects GDP growth (for example, from 2017 to 2019, the total domestic spending on the development of the digital economy increased from 3324 to 4094 billion rubles, and the contribution of the Russian Internet economy (Runet) to the GDP of Russia in 2021 was 9.5 trillion rubles, 42% higher than the year before), the Russian economy today is still heavily dependent on export revenues.

The second reason is the increase in the risk of destabilizing factors and threats, both external and internal in nature. Their impact can be assessed as particularly acute in the period of global economic recovery after the COVID-19 pandemic.

Despite the fact that the formation of the digital economy and the introduction of the results of scientific and technological progress in all spheres of economic life have positive economic effects, the development of high-tech industries abroad entails increased competition in world markets.

Under these conditions, the most important resource for the creation of domestic high-tech industries that can meet modern international market requirements, as well as for the formation and provision of mechanisms for sustainable socio-economic development, are human resources of innovative type, with a high level of professional knowledge and experience, as well as the ability to make informed and timely decisions.

In this regard, there is a need to improve approaches to human capital management in the context of the increasing role of information technology in all spheres of the economic life of society in order to ensure a high level of socio-economic security of the regions of the Russian Federation.

There have been numerous attempts to define constitutive elements of the concept of "socio-economic security of the region". However, they all have a common feature—an emphasis on individual aspects of the concept, which allows us to argue about their one-sided nature.

All approaches to define the content of "socio-economic security of the region" are united by the vagueness of the internal essence of the concept and the lack of emphasis on its substantive load. According to the results of the descriptive analysis, the socio-economic security of the region can be considered as a state of the region, described by certain parameters, as a condition for regional development, and as a characteristic of the region [1–3]. With this in mind, the multiplicity of approaches to defining the content of the concept of "socio-economic security of the region" becomes clear. Therefore, each of the existing approaches aimed at determining the scope of "socio-economic security of the region" concept is legitimate and determines the guidelines, vectors direction and nature of actions in its provision, which are appropriate and feasible within the chosen approach [4]. The incompleteness and limitation of the available methodological apparatus for assessing the socio-economic security of the region do not allow us to create adequate analytical support of measures and programs to strengthen it.

Drawing on the analysis and systematization of approaches that define socio-economic security, coupled with the analysis of the structure of socio-economic category and its relationship with the category of "human capital", we can conclude that human capital as a socio-economic system as a whole, or its individual structural elements in relationship with socio-economic security, can act in the following ways:

−  as a direction of state policy on the development of human potential formulated in the Economic Security Strategy of the Russian Federation until 2030;
−  as indicators and target areas of the policy goals stated in the Economic Security Strategy of the Russian Federation in reference to "Development of Human Potential" at the regional level;
−  as an object of management in the system of economic security of the region as a carrier of informal institutions, as well as a set of qualitative characteristics of the units (households, private enterprises, etc.);

— as a resource used in the process of implementing measures of socio-economic orientation in order to achieve economic security in the region; by the resource they mean personnel that ensure that the economic security management is fully functional in the region;

— as a factor affecting the level of socio-economic security of the region. The factor is determined as positive if the opportunities and/or conditions for the formation of a high level of economic security are analyzed through the indicators of qualitative or quantitative characteristics of the human capital of the region (for example, availability of education, health care, labor mobility, etc.). The factor is determined as negative if threats associated with unfavorable conditions of human capital development in the region or qualitative/quantitative characteristics of human capital (low level and quality of life, poverty, propensity to morbidity in the population, low availability of education and health care, etc.) are analyzed.

There is a close dialectical relationship between socio-economic security and the development of human capital: first, a person is a part of society and the main productive force of development; second, human capital is the main resource of socio-economic development; and third, human capital is an object of socio-economic security.

This connection is so close that the security of human capital is an integral part of the socio-economic security of society.

The relevance of our study is defined by the insufficient development of the theory of human capital in the context of the national economic security strategy of Russia, in addition to the lack of research perspectives and the pressing problems of human capital development in a knowledge economy.

The system of information exchange and accumulation of knowledge in regard to the problems of human capital development in the area of economic security of geographically adjacent and remote neighbours has not been created.

Studies into the problems of human capital development and the practice of implementing international experience in Russia are carried out fragmentarily, without elaboration of conceptual approaches and methodological recommendations for use in Russian economic conditions.

The above circumstances and unresolved problems defined the purpose of our research, which is to assess the level of socio-economic security of municipalities of the Voronezh region, taking into account the place and role of human capital, which in turn will allow the development of more specific mechanisms of socio-economic security management under the influence of modern challenges and threats.

## 2. Materials and Methods

The main approaches to the analysis of socio-economic security at the mesolevel formed the methodological basis of the study. Firstly, it is the contextual approach in which socio-economic security is recognized as a condition for regional development. The latter is viewed as the result of the harmonious combination of economic and social conditions that provides social guarantees, state obligations, and the comfort of living in the region on the basis of maintaining active economic activities by regional authorities within their competence. Secondly, it is the protective approach within the framework of which the state of socio-economic security of the region is formed. Its formation is affected by different threats posed to the regional socio-economic system and to the results achieved by the regional government in the area of security.

The proposed methodological framework for assessing the socio-economic security of the region provides for the joint use of functional and indicator-based approaches. When presenting the results, the following methods were also used: comparative; economic-mathematical (for processing the primary indicators to assess the socio-economic security of the Voronezh region); graphic (to visualize the results of the study); statistical (the selection of indicators to assess the level of socio-economic security indicators of the region), as well

as cartographic (for visualizing the level of socioeconomic security of municipal units of the region).

The empirical base of the study was formed by the data collected from the Federal State Statistics Service of the Voronezh region, as well as the results of research published in scientific publications, the official reports of Russian and foreign scientists, and in the official reports of the World Bank.

The constitution of the Russian Federation, the laws of the Russian Federation, resolutions of the Government of the Russian Federation and the Voronezh region, which define the socio-economic relations and the formal framework of the problem under consideration, provided the normative-legal basis of the study.

The information base for assessing the socio-economic security of the region was formed by an aggregate set of indicators determined by the principles of objectivity, practicality, comprehensiveness, unambiguity of interpretation, transparency, and reliability. The selected indicators to assess the socio-economic security of the region meet the requirements of heredity, reliability, measurability, and consistency. An important characteristic of the selected indicators is the ability to analyze them over time and to measure intra-regional differentiation. The question of man from the position of capital is primarily an economic one, although in modern science it is an interdisciplinary problem. Therefore, the discussion of how theoretical ideas about the concept of human capital might evolve and its role in the development of the economy and society as a whole should begin with economic research. Quite rightly, many analysts attribute the beginning of the idea of human capital to William Petty, who, using the category of "living human forces," focused on two important points: first, he viewed them as a factor in the growth of a country's wealth, and second, he was the first to attempt to determine the stock of these forces related to wages as a lifetime rent and the market rate of interest. These ideas, containing both conceptual and instrumental bases, have not lost their relevance today. In the 19th century, the accumulation of knowledge of human capital was provided by the works of the researchers, as presented in Table 1.

**Table 1.** Development of approaches to the analysis of human capital in the 19th century.

| Researchers | Main Work | Key Research Results |
|---|---|---|
| L.H. von Jakob (1758–1827) | The Rules of Popular Economy, or the Science of Popular Economy. -Halle, 1805. | In calculating the costs of hiring a hired worker, the concept of foregone income, characteristic of capital utilization analysis, was applied [5] |
| J.R. McCulloch (1789–1864) | The Principles of Political Economy/J.R. McCulloh. -Alex Murray and Son, 1870. | Based on the assumption of an analogy between physical and human capital, it was argued that human capital should have a rate of turnover that is consistent with investment in physical capital, and the interest rate [6] |
| N.W. Senior (1790–1864) | An Outline of the Science of Political. -New York. Farrar & Rincart, 1939. | The work links human ability and skill and the cost of acquiring them to future earnings [7] |
| L. Walras (1834–1910) | Elements of Pure Economics. -Homewood, Ill.: Rickard D. Irwin, 1954. | The idea is proposed that the price of any person is determined as with other capital goods [8] |
| J.D. MacLeod (1821–1902) | The Elements of Economics. Vol.11 New York: D. Appleton & Co. 1881. | It separates the productive person from the nonproductive person. Only the productive person is thought to be viewed in terms of fixed capital [9] |
| I.G. von Thünen (1783–1850) | Der isolierte Stadt. Vol.11, Part 11.; originally published in 1875. | The concept of capitalized productivity-enhancing costs was introduced and should be included in the stock of total capital [10] |

All of the authors listed in the table are of interest to this study, both by their methodological ideas and by their approaches to instrumental and measurable assessments of human capital.

The recent history of the formation and subsequent development of the concept of human capital in economic thought begins in 1960s, with such researchers as G. Becker [11,12], who revealed the essence of human capital and for the first time studied its structure; E. Denison [13], who developed the ideas of investing in human capital and proposed an approach to calculating the rate of return on investment; T. Schultz [14], who considered human capital as a productive factor capable of accumulation and reproduction; J. Minzer [15], who studied the issues of human capital profitability, having proposed a production function of wages and having highlighted factors influencing workers' incomes; and A. Toffler [16], who highlighted the features of the information age worker.

Studies appeared shortly after this that expanded the ideas about the structure of human capital and linked it with intellectual capital. These issues are covered in the works of J. Kendrick [17] and T. Stewart [18].

In the Russian socio-economic scientific literature, the issues of human capital have been analyzed only since the 1990s. Domestic researchers consider them from different positions and at different levels of manifestation. The general theoretical approaches are discussed in the works of M.M. Kritsky [19], L. Thurow [20], R.M. Nureyev [21], N.S. Zotkina, M.S. Gusarova, A.V. Kopytova [22], T.V. Chubarova [23], E.I. Kuznetsova, A.N. Osipova [24].

Human development in the socio-economic security system is a two-way process [25]. On the one hand, work is being done on the formation of human beings and their abilities and skills. The issues of labour, emotional-psychological, physiological and intellectual competence, and resilience are put in a single line.

On the other hand, sufficiently developed potential and the realization of acquired abilities are used for productive purposes in interaction with the external environment. Thus, another plane of intersection between the social and economic in the development of human potential is encountered.

The interaction of social and economic institutions through the system of social relations influences transforms human resources according to their own purposes.

In the process of activity, individuals influence social and economic structures, reproducing and changing them, while undergoing their own transformation. This determines the cyclicity of processes and the interconnection of subjects and objects of both human potential development and socio-economic security, where the environment forms human potential, and the potential realized in the process of activity reproduces and transforms this environment.

Man is the main subject and object of socio-economic security and human potential. That is, a person, a personality, simultaneously acts as both a producer and a consumer: on the one hand, as a subject who creates, forms, develops and realizes the potential and, when participating in economic activity, ensures a state of the economy in which its preservation and development will be possible; -on the other hand, as an object for which the state and institutions create favorable socio-economic conditions and the economic process is carried out.

At the same time, the actors who provide and distribute human potential, and the objects that are distributed and affected by human potential, in addition to the person himself, will be the state, its socio-economic institutions, and its organizations.

Despite the fact that economists and sociologists are currently showing great interest in the phenomenon of human capital, many theoretical and practical issues related to the definition of its role and place in the socio-economic security strategy of Russia remain unresolved.

The number of fundamental studies devoted to this problem is insignificant. This is explained, first, by the fact that the category of "human capital" is a relatively recent

subject of study in economic theory; and second, by insufficient attention to the categories associated with the person and their economic interests in Russia.

## 3. Results and Discussion

Issues of socio-economic security of the regions of Russia are of cardinal importance not only as a component within the scope of national security, but also in the context of the sustainable development of the state. The national level of socio-economic security is determined by the regional systems and the process of ensuring that it is implemented in compliance with the security of the territorial components. Monitoring socio-economic security at the regional level will therefore provide an opportunity to identify risks and develop mechanisms to guard against internal problems affecting the state functioning mechanism from within and disrupting the integrity of the entire socio-economic system.

Formally, the socio-economic security of a country can be represented as follows (Figure 1).

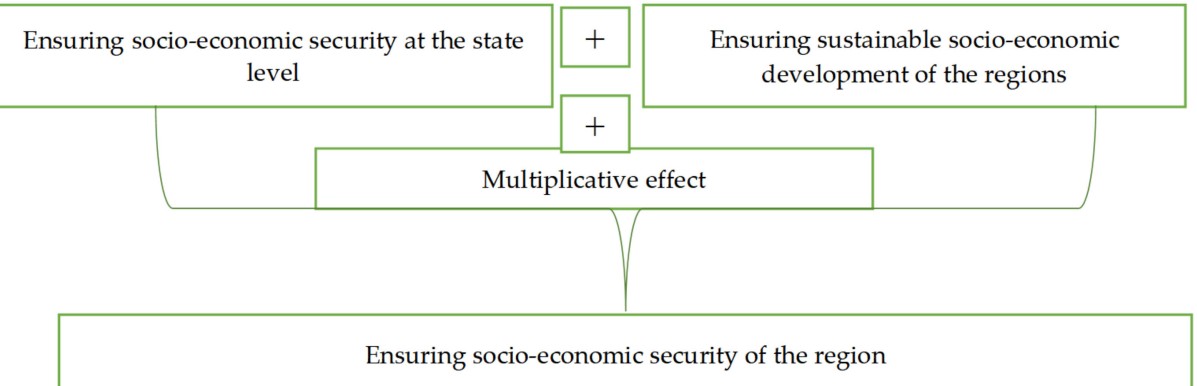

**Figure 1.** Ensuring the socio-economic security of the region.

The important thing at the information stage of the algorithm for assessing the socio-economic security of a region, the results of which decisively determine its assessment, is the formation of a set of indicators of socio-economic security of the region by which it is assessed. The ability of a set of indicators to capture all aspects of a particular component of socio-economic security in a region determines, among other things, the reliability of its intermediate assessments.

The calculations of both the integral assessment and the intermediate calculations for each allocated block are performed. The intermediate assessment of the socio-economic security of the region makes it possible to identify the components with the lowest assessments, i.e., to act as a basis for the reverse analysis, the purpose of which is to identify the causes of the unsatisfactory state of the parameters of socio-economic security of the region and the localization of the realized threats to the socio-economic system of the region. Low intermediate indicators of socio-economic security of the region are a direct consequence of the implementation of threats to the socio-economic system of the region which caused transformations of a negative nature in it.

At the first stage, the evaluation indicators of security of the socio-economic system of municipalities are selected in view of the goals and objectives set in the study. The selected evaluation indicators constitute three blocks of indicators—social, economic, and financial (Table 2).

**Table 2.** Indicators for assessing the socio-economic security of municipalities of the Voronezh region.

| Assessment Block | Indicator | Weighting Coefficient |
|---|---|---|
| Social | Birth rate | 0.2 |
| | Mortality rate | 0.2 |
| | The extent of paved roads to 10,000 km$^2$ | 0.1 |
| | Hospital beds per 1000 people | 0.1 |
| | Improvement of the housing stock—the specific weight of the total area of residential premises with basic infrastructure (water supply, drainage, heating, hot water, gas or electric stoves), | 0.1 |
| Economic | The volume of paid services per capita | 0.1 |
| | Commissioning of residential buildings in square meters per 1000 people | 0.1 |
| | The ratio of the total volume of retail trade and public catering per capita | 0.1 |
| | The volume of investments in fixed assets per capita | 0.4 |
| | The value of the average salary of employees of organizations | 0.2 |
| | The volume of agricultural production per capita | 0.3 |
| Financial | The share of tax and non-tax revenues in the total volume of local budget revenues, | 0.5 |
| | Average per capita budget security (local budget expenditures per 1 one resident) | 0.2 |
| | The share of profitable organizations | 0.3 |

It should be understood that the assignment of some indicators to one or another group is a conditional choice of the author, as such indicators can equally assess both the economic and social subsystem of the overall socio-economic system of the region [6–8]. Such indicators include the commissioning of residential buildings; the density of paved roads; and the value of the average wage of employees of organizations. At first sight, all of these indicators are purely economic indicators of regional economic development. However, on the other hand, income is the main means of livelihood and well-being of the population; in addition to new, affordable housing (the place of existence of the population and the creation of the primary unit of society); the availability of quality roads, which are an indicator of migration and mobility of the population, as well as rapid access to social services, medical care, and immediate help in emergency situations. Thus, these indicators can be used as indicators of the development of the social system of the subject of the study at the discretion of the authors. According to the impact on the socio-economic system, indicators can be classified as both positive and negative. When an increase in any indicator leads to the deterioration of the socio-economic system, i.e., the growth of the indicator has negative consequences, such indicators are destabilizing. In the given methodology, the death rate is a destabilizing indicator. The growth of other indicators has a positive effect, i.e., it stimulates the development of the socio-economic system. Furthermore, when developing a methodology for assessing the differentiation of development, it is necessary to take this fact into account and to apply various formulas to convert them into integral coefficients. Currently, there are many popular and important social and economic indicators used in the assessment of the socio-economic situation of the region as a whole. For example, the fundamental economic indicator is the volume of goods and services shipped in different areas of production. Such an indicator is very important for assessing differentiation, but at a higher level, the federal one, since not all municipalities of the region are represented by one or another sphere of production, or there are fewer than three such enterprises.

The calculation methodology involves a number of sequential steps. To begin with, we convert all of the different dimension indicators into a single integral form.

$$X_{ip} = \frac{A_{ip}}{A_{i\,\text{cp}}} \tag{1}$$

where:

$X_{ip}$—integral indicator—*i*, municipality—*p*;
$A_{ip}$—absolute indicator—*i*, municipality—*p*;
$A_{i\text{ cp}}$—regional average indicator—*i*.

Further, the integral indicators are converted into points by means of a ball score, the system of which is presented in Table 3.

**Table 3.** The system of score evaluation of integral indicators of socio-economic security of municipalities.

| Indicator | Points | | | |
|---|---|---|---|---|
| | **0** | **0.5** | **1.0** | **1.5** |
| Positive | $X_{ip} < 0.5$ | $0.5 \leq X_{ip} < 1$ | $1 \leq X_{ip} < 1.5$ | $X_{ip} \geq 1.5$ |
| Negative | $X_{ip} \geq 1.5$ | $1 \leq X_{ip} < 1.5$ | $0.5 \leq X_{ip} < 1$ | $X_{ip} < 0.5$ |

Each indicator is then given a weighting coefficient depending on the impact on the security of the socio-economic system of municipalities (Table 1). The final integral indicator of socio-economic security is presented as the sum of 14 weighted integral indicators for each municipality.

$$V_p = \sum_{i=1}^{n} (k_i \times x_{ip}) \tag{2}$$

where:

$V_p$—the final integral indicator of socio-economic security of the municipality—*p*;
$x_{ip}$—integral indicator—*i*, municipality—*p*;
$k_i$—indicator weighting coefficient—*i*.

In the final step, the number of clusters and the threshold values (group interval) of the integral indicator of the region's sustainable development should be determined. In our case, we propose to distinguish five groups of municipalities according to the level of socio-economic security: high, medium, below average, low, and crisis.

Thus, the presented methodology of assessing the level of socio-economic security of the region, implemented using Microsoft Excel, includes the following steps:

(1) sampling and entering absolute values of static indicators into the program;
(2) calculating indices of growth of indicators (each subsequent year to the previous year), a dynamic indicator for the period;
(3) developing criteria for assessing static and dynamic indicators on a three-point scale based on a comparison with the average level of indicators in the region;
(4) the quantitative (point) assessment of indicators in accordance with the system of criteria;
(5) determining the weight of indicators within the block of indicators (indicators are an assumed equilibrium, but their weight is differentiated according to the number of indicators within the block. Each of the four blocks of indicators is given a weight of 25% of the total score);
(6) weighing the evaluation of the value of the indicators by multiplying the score by the weight of the indicator;
(7) determining the final assessment of the indicator by summing up the assessments. In order to make a comparative assessment of the level of socio-economic security of the regions, their rating is compiled by sorting the values of the assessment of the level of economic security by the criterion of decreasing values (starting with the maximum and ending with the minimum values).

This methodology is tested on the example of the Voronezh region using statistical data for 2019. The results of the research are presented in the form of a map (Figure 2).

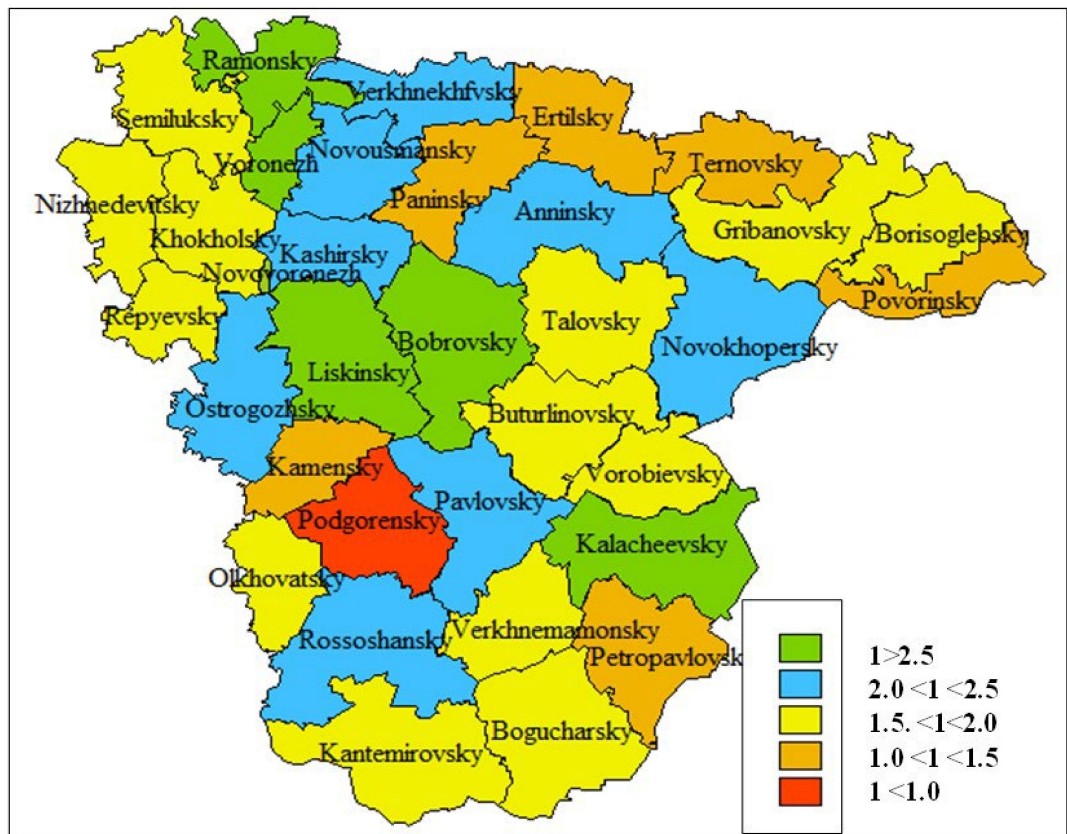

**Figure 2.** Integrated index of social and economic security (compiled by the author).

The first group with a high level of socio-economic security includes municipalities in the north of the region, including the Voronezh city district and Ramonsky district; Novovoronezh city and the Liskinsky and Bobrovsky districts in the center; and the Kalacheevsky district in the south. These municipalities are characterized by above-average indicators for all statistical parameters and are the "engines" of socio-economic development of the entire Voronezh region. The eight municipalities represent the group with an average level of socio-economic security. They are above the regional average by most parameters. Territorially, they are mainly located in the central part and are directly adjacent to municipalities with favourable economic conditions. The largest group by number of subjects (13) is the group with below average indicators (unstable conditions). These municipalities can perform well in some areas, but also negatively in others. Their territorial position: peripheral to the west, east and south of the region, respectively, is noteworthy. The group posing the risk of socio-economic security (the level 'below average') includes six districts of the oblast: Petropavlovsky in the south, Kamensky in the centre, Paninsky, Ertilsky, Ternovsky in the north, and Povorinsky in the east. If local and regional authorities do not act soon to improve the socio-economic situation, this will increase the migration outflow of the populations from these territories to the regional centre to a critical level, which, in turn, will call into question the very existence of these municipalities. In 2019, the Podgorensky district showed the worst result out of all 34 municipalities in almost all indicators and, as a consequence, the worst integral indicator of socio-economic security.

This district belongs to the group with a crisis level of socio-economic security, which is due to its underdeveloped economic base, lack of industrial enterprises, and jobs, as well as low investment activity. These problems require radical decisions by the leadership of the region, the development of individual programmes to support entrepreneurs and small and medium-sized businesses, and to attract investment in a particular area. The calculations on the integral index of socio-economic security show that there are significant socio-economic disparities between the municipalities of the Voronezh region which require the

adoption of radical measures at the regional level. The content of regional development will contribute to the elimination of imbalances in the municipal economies, the development of stabilization processes in the socio-economic life of the region, and, accordingly, improve the quality of human potential. Regional imbalances as a natural process of socio-economic development can be redressed in certain ways, such as by developing local self-government in economic and social parameters of functioning of settlements of different taxonomic levels. Inter-municipal relations are an essential lever of dynamically balanced socio-economic, scientific, technical and cultural development of both individual municipalities and the region as a whole.

The defining priority for the region should be the development and implementation of a regional programme aimed at designing an innovative model of socio-economic growth, establishing the region as a high-tech area. The task must be implemented through a new strategy of innovation, the introduction of a reliable system of resource conservation, low-waste production, and the implementation of innovative technologies. The structural, fiscal, financial, macroeconomic and regional policies of the government and other executive bodies should be aimed at this. The prerequisite for the success of such policies at all levels is a fundamental management reform.

The competence of municipalities covers structural changes in labour markets, private sector development, and the introduction of foreign investment into the region. The leading socio-economic factors are the availability of highly qualified personnel and the existence of scientific centres and educational institutions, which contributes to the development of scientific and technological progress in the region. A strong and stable regional economy, and consequently the socio-economic security of municipalities, should be formed on the basis of the potential that municipalities have in regard to resource development, the stimulation of a more complete and effective use of local labour, and intellectual and other resources.

## 4. Conclusions

Overall, it is clear that positive economic growth is made possible by the positive growth of human capital, which, in turn, contributes to the development of the country's innovation potential. Today, the difference in the growth rates of countries is due to the differences in the number of developed and developing countries standing at different levels of human capital development. The difference in growth rates between depressed and developed municipalities in the Voronezh region is due to the fact that more developed municipalities have higher levels of productivity and consequently higher levels of human capital quality. Municipalities that need to increase their productivity to expand the new qualities of their human capital now need to adopt industrial and technological innovation, as encouraged by both business stakeholders and politicians. It should be noted that socio-economic growth in the region will be feasible on the condition that there are widespread and comprehensive innovations at all stages of its reproductive process.

The generator and consumer of all new ideas is the person (citizen and customer), who is also a highly qualified employee with up-to-date knowledge (which must be constantly updated and supplemented throughout his/her life), as well as real life experience. Ensuring the socio-economic security of the region is possible through the functioning of an effective organizational and economic mechanism, represented as an interrelated system of economic relations within the overall economic system. Methods of economic management include planning, financing, motivation, accounting, and analysis, while organizational methods include regulation and management, organizational design, decision-making methods, and problem-setting and control methods.

Programmed-target programming allows for the formulation of regional socio-economic policy priorities; the determination of the sequence and timing for solving the identified socio-economic problems; the stabilization of business conditions; increased business activity; the harmonization of the activities of economic entities; the rational integration of the interests of industries and individual territories aimed at developing the entire regional

socio-economic system and, consequently, the provision of an acceptable level of social and economic development of the region. An assessment of the indicators of socio-economic security of the region has made it possible to determine the following regularities: (1) the regional level of socio-economic security is significantly affected by both national trends in different spheres of the society in addition to global trends; (2) socio-economic security in the region is provided mainly by the processing industries; and (3) the Voronezh region has a certain potential for development, as it has a variety of natural and human resources.

Socio-economic security is a synthetic category containing economic and social components. At the same time, it is a universal category that is applicable to all levels of the socio-economic system of the state. The objects of socio-economic security are the state, the region, and the economic entity (enterprise). For each of these objects, socio-economic security is an important condition for their operation and development. At the same time, each object of socio-economic security has certain features (in the assessment and provision) due to the characteristic features of the object. Each of these objects has its own peculiarities in development and, accordingly, the assessment tools are different for them. These peculiarities are due to the characteristics of the socio-economic security object (individual, economic entity, city, region and state), which act as the basis for the creation of the socio-economic security base of the corresponding level. To summarize the above, the socio-economic security of the Voronezh region presupposes the development of all its constituent territorial units. The results of the study can be used in the plans and strategies for the development of the region as a whole, and each municipality in particular, for the medium- and long-term. It should be emphasized that the region is, a priori, heterogeneous in terms of socio-economic development, and it is not possible to achieve "perfect" homogeneity. Nevertheless, smoothing out territorial disparities as much as possible in the given circumstances is one of the fundamental objectives of regional policy and local government.

The results, generalizations and conclusions obtained by the authors clarify the categories of "socio-economic security" and "human capital".

The practical recommendations offered by our research can be useful in for creating the concept of regional socio-economic security, regional programs of socio-economic development, and the elaboration of specific proposals for the development of human capital in the Voronezh region.

The identified trends in the level of socio-economic security of the region can be used by public authorities to ensure the stable and sustainable development of the regional economy and its human capital.

Thus, from the point of view of the system approach, the elements of the category of "human capital", as a complex multidimensional structure, can be present at different levels of the system of socio-economic security of the region.

In this regard, we can conclude that the formation of a conceptual model of socio-economic security system of the region, taking into account the place and role of human capital, will allow the development of more specific mechanisms for the management of socio-economic security affected by modern challenges and threats.

The formation of a mechanism that will ensure the socio-economic security of the country is impossible without its comprehensive assessment by a number of criteria.

The formation of a mechanism to ensure economic security, taking into account all the substantive determinants of socio-economic security of the region, will contribute to the definition of targets for achieving the safe development of the state, and will lead to the next, qualitatively new stage of development, where security can be defined as a certain state of socio-economic development of the country, The latter is characterized by the most complete and rational use of its socio-economic potential through the lens of regional interests.

The state policy of regional development entails changes in the internal socio-economic structure of the region and is reflected both in the relationship between the state and the regions and in the relationships between the regions themselves.

The state policy of regional development is based on the distinctive characteristics of the regions, the formation of conditions for a more effective and harmonious development of the regions, as well as on improving the quality of life of the states' population.

This purpose will be achieved through the implementation of a state regional policy aimed at developing each region's potential, overcoming infrastructural and institutional limitations, creating equal opportunities for citizens and promoting human development, carrying out targeted work to develop federal relations, and reforming public administration and local self-governance systems.

**Author Contributions:** Conceptualization, N.V.Y. and L.S.; data curation, M.Y.T.; formal analysis, G.I.Z.; funding acquisition, E.A.S.; methodology, N.V.Y.; software, I.A.B. and E.A.S.; writing—original draft, N.V.Y.; writing—review and editing, N.V.Y. All authors have read and agreed to the published version of the manuscript.

**Funding:** The reported study was funded by RFBR (Project no. 19-29-07400-mk).

**Institutional Review Board Statement:** Not applicable.

**Informed Consent Statement:** Not applicable.

**Data Availability Statement:** Not applicable.

**Conflicts of Interest:** The authors declare that they have no conflict of interest.

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
