# Peer review of "Socio-Economic Security of the Region in the Context of Human Capital Development"

_sustainability, doi:10.3390/su15010404_

Round 1

Reviewer 1 Report

This paper aims to discuss the concept of socio-economic security and to define a measurement method related to human capital. The authors mapped data from the public statistical service and GIS data.

The research question is undoubtedly attractive, but the paper must be improved.

·         Originality/Novelty: Is the question original and well defined? Do the results provide an advance in current knowledge?

The question is not entirely original. Anyway, the results provide a particular advance in current knowledge. 

·          Significance: Are the results interpreted appropriately? Are they significant? Are all conclusions justified and supported by the results? Are hypotheses and speculations carefully identified as such?

Results are commented on properly; hypotheses are correctly identified; the Basic conceptual model presented in the paper is based on good literature.

·          Quality of Presentation: Is the article written in an appropriate way? Are the data and analyses presented appropriately? Are the highest standards for presentation of the results used?

The article is written in Standard English. Some parts must be clarified.

·         Scientific Soundness: is the study correctly designed and technically sound? Are the analyses performed with the highest technical standards? Are the data robust enough to draw the conclusions? Are the methods, tools, software, and reagents described with sufficient details to allow another researcher to reproduce the results?

The study seems correctly designed, methods used are well-known in the literature.

·          Interest to the Readers: Are the conclusions interesting for the readership of the Journal? Will the paper attract a wide readership, or be of interest only to a limited number of people? (please see the Aims and Scope of the Journal)

The conclusion could be satisfying for the readership of the Journal.

·          Overall Merit: Is there an overall benefit to publishing this work? Does the work provide an advance towards the current knowledge? Do the authors have addressed an important long-standing question with smart experiments?

Yes.

·         English Level: Is the English language appropriate and understandable?

The article is written in standard English, but there are some imperfections. Some parts must be reviewed.

Review Comments to the Author:

Dear Authors, thanks for submitting a contribution to this field of study. I have several considerations for expansion and clarification:

- First, as I wrote upward, I suggest you perform a language check to enhance the overall readability.

- I find the introduction not clear. Assumptions concerning globalization aren’t based on good literature, and I think you should discuss more effects. I think you should consider rewriting the entire section.

- Anyway, the literature part in the second chapter is excellent; I would consider shifting it to the first section.

- I find your conclusions interesting and properly founded on data; I hope the political stakeholders will consider your research.

Author Response

we tried to take into account all the comments. all changes are shown in color in the corrected version

Reviewer 2 Report

The article discusses an interesting and topical issue. However, small changes need to be made: 

Firstly, the introduction does not adequately define the objectives pursued by the research. 

There is no Theoretical Framework section detailing the theoretical background on which the hypotheses of this article have been based. Therefore, this section should be included.

In the second section, Materials and Methods, it does not explain well why this methodology was chosen. Would it be possible to introduce any added advantage that this method provides? 

Nor is there a Discussion section that discusses the results you have obtained with those of other previously published studies. Therefore, it would be advisable to include this section. 

Finally, the conclusion does not describe in detail what it contributes to the literature, nor does it describe future lines of research. 

Author Response

(The authors gave the same response as above.)

Reviewer 3 Report

The proposed topic is very interesting especially that it approaches a highly debated topic: socio-economic security. 

However, the purpose of the study is not clear. Moreover, the literature review part is not properly presented. The authors should have coherently argued the findings of other studies and, based on them, they should have developed the research hypothesis.  

Regarding the methodological part, the authors have conducted only a descriptive research, without making a solid statistical analysis. 

The paper does not present sound and relevant policy implications.

The paper needs proof-reading.

Author Response

(The authors gave the same response as above.)

Round 2

Reviewer 1 Report

Dear Authors,

many thanks for this revision and your comments!

I think that you positively responded to my comments, and now both the introduction and the conclusions are properly written.

Nevertheless, I think that the paper still needs proofreading to be published.

Author Response

The article has been revised in accordance with the wishes of the reviewer. Many thanks for the valuable comments. The interpreter fixed everything. The changes are in red.
